# Content and Mechanism of Action of National Antimicrobial Stewardship Interventions on Management of Respiratory Tract Infections in Primary and Community Care

**DOI:** 10.3390/antibiotics9080512

**Published:** 2020-08-13

**Authors:** Lou Atkins, Tim Chadborn, Paulina Bondaronek, Diane Ashiru-Oredope, Elizabeth Beech, Natalie Herd, Victoria de La Morinière, Marta González-Iraizoz, Susan Hopkins, Cliodna McNulty, Anna Sallis

**Affiliations:** 1Center for Behavior Change, University College London, Alexandra House, 7–19 Queens Square, London WC1N 3AZ, UK; natalie.herd@gmail.com; 2Public Health England Behavioral Insights (PHEBI), Public Health England, Wellington House, 133–155 Waterloo Road, London SE1 8UG, UK; tim.chadborn@phe.gov.uk (T.C.); paulina.bondaronek@phe.gov.uk (P.B.); diane.ashiru-oredope@phe.gov.uk (D.A.-O.); M.Gonzalez-Iraizoz@warwick.ac.uk (M.G.-I.); susan.hopkins@phe.gov.uk (S.H.); anna.sallis@phe.gov.uk (A.S.); 3NHS England and NHS Improvement, Wellington House, 1st Floor, 133-155 Waterloo Road, London SE1 8UG, UK; elizabeth.beech@nhs.net; 4Chelsea and Westminster Hospital NHS Foundation Trust, London SW10 9NH, UK; Victoria.delamoriniere@chelwest.nhs.uk; 5Public Health England, Primary Care Unit, Twyver House, Bruton Way, Gloucester GL1 1DQ, UK; Cliodna.McNulty@phe.gov.uk

**Keywords:** antimicrobial stewardship, primary care, behavior change wheel, behavior change techniques, theoretical domains framework, respiratory tract infection, RTI

## Abstract

A major modifiable factor contributing to antimicrobial resistance (AMR) is inappropriate use and overuse of antimicrobials, such as antibiotics. This study aimed to describe the content and mechanism of action of antimicrobial stewardship (AMS) interventions to improve appropriate antibiotic use for respiratory tract infections (RTI) in primary and community care. This study also aimed to describe who these interventions were aimed at and the specific behaviors targeted for change. Evidence-based guidelines, peer-review publications, and infection experts were consulted to identify behaviors relevant to AMS for RTI in primary care and interventions to target these behaviors. Behavior change tools were used to describe the content of interventions. Theoretical frameworks were used to describe mechanisms of action. A total of 32 behaviors targeting six different groups were identified (patients; prescribers; community pharmacists; providers; commissioners; providers and commissioners). Thirty-nine interventions targeting the behaviors were identified (patients = 15, prescribers = 22, community pharmacy staff = 8, providers = 18, and commissioners = 18). Interventions targeted a mean of 5.8 behaviors (range 1–27). Influences on behavior most frequently targeted by interventions were psychological capability (knowledge and skills); reflective motivation (beliefs about consequences, intentions, social/professional role and identity); and physical opportunity (environmental context and resources). Interventions were most commonly characterized as achieving change by training, enabling, or educating and were delivered mainly through guidelines, service provision, and communications & marketing. Interventions included a mean of four Behavior Change Techniques (BCTs) (range 1–14). We identified little intervention content targeting automatic motivation and social opportunity influences on behavior. The majority of interventions focussed on education and training, which target knowledge and skills though the provision of instructions on how to perform a behavior and information about health consequences. Interventions could be refined with the inclusion of relevant BCTs, such as goal-setting and action planning (identified in only a few interventions), to translate instruction on how to perform a behavior into action. This study provides a platform to refine content and plan evaluation of antimicrobial stewardship interventions.

## 1. Introduction

The number of serious infections resistant to treatment is increasing and antimicrobial resistance (AMR) is one of the major risks facing public health [1,2]. In Europe, AMR is associated with approximately 25,000 deaths per year and 700,000 globally [3]. It is estimated that a continued rise in resistance would cost the world 100 trillion USD by 2050 if AMR is not addressed effectively [4]. The Interagency Coordination Group on Antimicrobial Resistance report to the WHO recommends countries reduce the need for antimicrobials and enhance their responsible and prudent use, as well as advises the use of behavior change interventions aimed at both public and professionals [5].

One of the major modifiable factors contributing to AMR is inappropriate use and overuse of antimicrobials such as antibiotics [3]. In the UK, 72% of antibiotics are prescribed in General Practice (GP) [6]. Although consumption of antibiotics in UK primary care decreased by 16.7% between 2014–2018 [6], this more than a third higher than some other European countries, such as Sweden and The Netherlands [7].

The first step in intervention design is to specify the behavior(s) the intervention is aimed at changing [8]. Both prescribers’ and patients’ behavior is associated with the inappropriate use and misuse of antibiotics [9]. Prescribers may issue unnecessary prescription of antibiotics to patients with self-limiting infections [10] or inappropriately select the type and duration of the medication [11]. A European Center for Disease Prevention and Control survey of healthcare professionals reported that only 63% agreed or strongly agreed they have a key role in helping control antibiotic resistance [12]. Public misconceptions on the indications and effectiveness of antibiotics are prevalent with only 43% of respondents in a European Commission survey of general public views of antimicrobial resistance knowing that antibiotics are ineffective against viruses [13]. Patient’s expectations around the use of antibiotics can influence the GP’s prescribing behavior and lead to inappropriate prescribing and overuse [14].

The optimization of prescribing practice through antimicrobial stewardship (AMS) programs was one of seven key priority areas for action in the UK Five Year Antimicrobial Resistance Strategy 2013 to 2018 [15]. AMS includes various initiatives to tackle AMR and improve patient safety, for example: establishing optimal standards for antimicrobial use within the healthcare setting, interventions aiming to promote appropriate prescribing, and reviewing the impact of the AMS initiatives. Multifaceted AMS programs aimed at the public, as well as frontline healthcare professionals, are needed to tackle AMR [16,17]. Moreover, there is evidence that the culture of the healthcare organization may also influence the prescribing behavior [18].

A number of policies and interventions have been developed and nationally implemented. However, it is unclear if these interventions focus on the behaviors that would optimally impact on AMR and whether they use the full range of intervention types and policy levers available. It is also unclear if the current program of AMS interventions aim to change behavior through the mechanisms hypothesized to be effective in behavior change theory.

Pinder et al. [19] conducted a review of 150 studies and identified key behaviors that should be targeted in AMS interventions and proposed potential opportunities for new interventions. They also identified a small number of nationally implemented interventions that were mapped to the barriers and facilitators found in the literature. To optimize the potential of AMS programs, the behaviors and populations targeted by these interventions, as well as their content and mechanisms of action of these needs, to be articulated. Tools developed in behavioral science can support such work.

The Behavior Change Wheel (BCW) [20] is a synthesis of 19 frameworks of behavior change and can be used to characterize the content of interventions. The BCW is linked to a model of behavior, COM-B (Capability, Opportunity, Motivation–Behavior), which can be used to identify the mechanisms of action of interventions. The Theoretical Domains Framework (TDF) can be considered an elaboration of COM-B (see Appendix A). McParland [21] used the TDF and the Behavior Change Technique (BCT) Taxonomy version 1 (BCTTv1) [22], a classification of “active ingredients” that are used to change behavior, to describe public awareness AMR interventions. The authors concluded that there is a clear potential for improvement of interventions. However, the review focused on 20 peer-reviewed studies of interventions aimed at the general public only. There remains a need to review nationally implemented interventions that target both healthcare professionals, as well as the public. To the authors’ knowledge, the BCW has not been used to review AMS interventions already implemented nationally across England to analyze and improve the AMS initiatives.

The overarching aim of this study was to build on the work of Pinder et al. [19] by characterizing nationally implemented AMS primary care programs to identify any gaps in coverage and opportunities for refinement. The specific aims were to:identify the behaviors and target populations related to AMS for respiratory tract infection (RTI) in primary care;identify interventions targeting AMS for RTI in primary care;describe their content using the Behavior Change Wheel and BCT;describe their mechanisms of action of interventions using COM-B and TDF.

Research identifying gaps between the influences and behavioral content of AMS interventions to highlight potential avenues for improvement will be reported in a separate study.

## 2. Methods

We drew on the following sources to identify behaviors relevant to patient self-care and/or appropriate antibiotic advice for the management of signs and symptoms of self-limiting respiratory tract infections:A literature review and high level behavioral analysis of antibiotic prescribing in the public, primary and secondary care [19].Relevant official guidance [17,23,24,25,26,27,28,29,30,31,32].UK AMR five-year strategy [15].Consultation with experts in epidemiology, pharmacy, infection management, and behavioral science.Self-limiting respiratory tract infections and symptoms of respiratory tract infections, included acute cough/acute bronchitis, common colds, flu, acute otitis media, acute otitis externa, (middle) ear infections, acute sinusitis, sore throat (tonsillitis), pharyngitis, laryngitis, bronchitis, respiratory tract infection, tracheitis, and acute rhinitis.

To describe the content of nationally adopted AMS interventions in England, authors identified potentially relevant programs for inclusion based on their knowledge of the policy area, consultation with key stakeholders and review of relevant documentation.

We included interventions implemented nationally in England between January 2014 and February 2018, where the primary objective was antimicrobial stewardship activities related to patient self-care and/or appropriate antibiotic advice for the management of respiratory tract infections. Interventions were excluded if local implementation only had occurred. Two authors (AS and VdLM) sought descriptions of the interventions from publicly available material and in some instances contacted the program leads for elaboration and to ensure accurate descriptions.

Providers and commissioners were defined as follows: providers defined as professionals in organizations providing care for the management of signs and symptoms of RTIs, e.g., Trusts, General Practice, private providers, and commissioners defined as professionals in organizations commissioning services for the management of signs and symptoms of self-limiting RTIs, e.g., Local Authorities, Clinical Commissioning Groups, Clinical Services Units, and National Health Service England.

### Data Extraction Tools

The BCW and BCTTv1 were used to classify intervention content identified in intervention materials or descriptions of interventions into intervention types, policy options (see Appendix A), and BCTs.

To describe the intervention mechanisms of action, we were guided by three matrices—two linking BCTs to TDF domains [33,34] and one linking intervention types to COM-B and TDF [8] to determine from the BCTs we identified in interventions, which influences on behavior (characterized using COM-B and TDF) the BCTs were likely to be targeting.

## 3. Data Analysis

For each intervention, we recorded the total number of COM-B components and TDF domains, intervention types, policy options, and BCTs, and calculated the mean and range. We recorded the number of interventions in which each COM-B component, TDF domain, intervention types, BCT, and policy option was present (mean and range).

### Interrater Reliability

One researcher (LA) conducted 100% of the coding, and a second researcher (AS) coded 20% to establish reliability. Percentage agreement for each section of intervention coded was calculated where both coders assigned the same code(s) to the same section of intervention material. PABAK (Prevalence And Bias-Adjusted Kappa) statistic [35] was used to assess the presence or absence of codes within each intervention. This statistic adjusts for shared bias in the coders’ use of options and high prevalence of negative agreement, i.e., when both coders agree that codes are not present. Interrater reliability of 0.60–0.79 = ‘substantial’ reliability, >0.80 = ‘outstanding’ [36].

## 4. Results

We identified 32 behaviors related to AMS for RTI in primary care. These are summarized in Table 1 (see Appendix A for the sources from which behaviors were identified). Five were patient behaviors, 11 related to prescribers, five were community pharmacist behaviors, two related to providers, one related to commissioners), and eight related to both providers and commissioners.

A total of 83 interventions were identified for content analysis of nationally adopted AMS interventions in England. Thirty-nine met the inclusion criteria (Table 2 with the target group(s)). Of the 44 excluded interventions: 31 were not aimed at the relevant behaviors or target groups; five were information, general media, or links to resources only; four were not nationally implemented; and four were part of other already identified interventions.

Of the 39 included interventions, 15 were aimed at patients/public, 22 at prescribers, eight at community pharmacy staff, 18 at providers, and 18 at commissioners. Almost half of the interventions (n = 19) were aimed at one target group only, nine were aimed at two target groups and three interventions (Public Health England’s Antibiotic Guardian [37], TARGET Antibiotics Toolkit (Treat Antibiotics Responsibly, Guidance, Education, Tools) [38], and British Society for Antimicrobial Chemotherapy’s Antibiotic Action [39]) were aimed at all five target groups. A mean of 5.8 behaviors were targeted by interventions.

Patient interventions were largely aimed at encouraging self-care and reducing requests for antibiotics for self-limiting respiratory tract infections (e.g., Antibiotic Guardian, Treat Yourself Better [40]). Interventions aimed at prescribers targeted various behaviors, but most commonly the behavior that antibiotic prescriptions ‘should be written only when there is a clear clinical benefit’ (e.g., UK Department of Health and Public Health England’s Antimicrobial Prescribing and Stewardship Competencies [41]), followed by the behavior to ‘provide a backup prescription where appropriate’ (e.g., TARGET). The behavior targeted least was that prescribing antibiotics following a telephone consultation should only occur in exceptional circumstances (e.g., NICE Infection Prevention and Control (QS61) [27]). Interventions targeting pharmacies were largely aimed at provision of self-care advice (e.g., NICE Antimicrobial stewardship: changing risk-related behaviors in the general population NICE guideline (NG63) [23]) and sharing written resources with the patient (e.g., The Learning Pharmacy [42]). The most frequently targeted behavior for providers and commissioners was ‘monitor antibiotic prescribing in relation to local and national resistance patterns or targets’ (e.g., TARGET).

### 4.1. Interrater Reliability

The kappa ranged from 0.60 to 0.89 suggesting good to very good inter-rater reliability. The PABAK was from 0.75 to 0.95, suggesting substantial to outstanding agreement. Kappa and PABAK for behaviors, mechanism of action (COM-B, TDF), and intervention content (intervention types, policy options, BCTs) are presented in Table 3.

### 4.2. Intervention Types Identified in Interventions

Eight of a potential nine intervention types were identified across all interventions (Table 4). The mean number of intervention types per intervention was 3 (range 1–6). Figure 1 shows the number of types identified in interventions split by target group. Patient interventions often used education, training and persuasion. Prescriber interventions included seven out of nine intervention types, primarily training, education, and persuasion. Pharmacy interventions frequently used enablement and training but also used education and persuasion covering four of the nine possible intervention types. Provider and commissioner interventions were most frequently training, enablement, and education. No interventions used restriction.

### 4.3. Policy Options Identified in Interventions

Five policy options were identified across the 39 interventions. ‘Service provision,’ e.g., website, was the most frequently identified (15/39), followed by ‘guidelines,’ e.g., NICE guidance, (14/39), ‘communication / marketing,’ e.g., poster, (10/39), ‘legislation,’ i.e., Health and Social Care Act (1/39), and ‘fiscal measures’, i.e., NHS Quality Premiums for CCGs (1/39). Figure 2 shows the number of policy options identified in interventions split by target group. No interventions were delivered through environmental/social planning or regulation. Patient interventions were commonly delivered using communication/marketing and service provision, while prescriber interventions were delivered mainly through guidelines, followed by service provision. Provider and commissioner interventions were delivered largely through guidelines and service provision, with some communication/marketing and legislation.

### 4.4. BCTs Identified in Interventions

A total of 30 BCTs were identified across all interventions with the mean number of 4 per intervention (range 1–14) as shown in Table 5. Figure 3 shows the number of BCTs identified in interventions split by target group. Patient and prescriber interventions most commonly used the BCTs ‘information about health consequences’ and ‘instruction on how to perform the behavior’. The most frequently used BCT in pharmacy, provider, and commissioner interventions was ‘instruction on how to perform the behavior.’

The frequency of mechanisms of action identified in interventions is outlined in Table 6.

Figure 4 and Figure 5 show the influences on behavior targeted in interventions split by target groups. The most frequently targeted influence on behavior was psychological capability for all target groups. For patients, prescribers and commissioners reflective motivation was the next most frequently targeted influence on behavior, whilst for pharmacy and provider interventions, it was physical opportunity. Automatic motivation and social opportunity were infrequently targeted. Only one intervention aimed to change physical capability.

The TDF domain, knowledge, was the most frequently targeted influence for all target groups; for pharmacy interventions, ‘environmental context and resources’ was equally prevalent. For patients and prescribers, the next most commonly targeted influences were beliefs about consequences and skills. For pharmacy it was social and professional role and identity and skills. Interventions for providers and commissioners also targeted behavioral regulation and environmental context and resources.

A summary of interventions, the behaviors and groups they target, and their content and mechanism of action is given in Table A1. The same information structured by behavior is provided in Appendix A.

## 5. Discussion

The overall aim of current research was to gain a detailed understanding of existing AMS national interventions, to pinpoint key areas of focus, and potential gaps and opportunities for future interventions. This study identified behaviors and target populations related to AMS for RTI in primary care and characterized the content and mechanism of action of interventions of nationally implemented AMS interventions targeted in primary care (patients and public, prescribers, providers, commissioners, and pharmacy).

Of the 32 behaviors identified, approximately one-third were related to prescribers, one third related both to providers and commissioners, and the remaining third was evenly split between patients and community pharmacy staff. The focus of interventions for patients was on self-care and not requesting antibiotics at consultations. The majority of prescriber interventions encouraged ‘prescribing only when there is a clear clinical benefit’, ‘giving alternative self-care advice’, ‘providing a back-up prescription where appropriate’, and ‘following local antibiotic formularies.’ Few interventions addressed limiting prescribing following telephone consultation and undertaking point of care tests (POCT), such as CRP. Given the potential for POCT to reduce inappropriate prescribing [67] related behaviors should be considered in the design or refinement of future interventions. Pharmacy interventions were aimed at provision of self-care advice, sharing written resources and checking antibiotic prescriptions comply with local guidance and querying those that do not. Provider and commissioner interventions focused on ‘monitoring prescribing in relation to local and national resistance patterns’, and ‘commissioning, developing, and implementing interventions to support AMS’.

We identified 39 national AMS interventions for RTI in primary care. Interventions were typically in the form of ‘service provision’, such as a website or clinical guidelines followed by communications/marketing. Approximately three-quarters of the interventions were ‘training’ mainly using the BCT, ‘instruction on how to perform the behavior’ and includes training programs, such as TARGET [38], Stemming the Tide of Antibiotic Resistance (STAR) e-learning [44], Managing Acute Respiratory Tract Infections (MARTI) e-learning [45], and Public Health England’s ‘Beat the Bugs’ course [46]. The same proportion served were classed as ‘education’ mostly frequently using the BCT ‘information about health consequences.’ Twenty-five interventions were classed as ‘enabling,’ delivered most frequently with the BCT ’adding objects to the environment,’ such as the provision of a checklist to prevent antimicrobial misuse. Eighteen interventions used ‘persuasion’ mostly through the BCT ‘credible source,’ for example, providing an experienced GP’s view of implementing delayed prescriptions. Few to no interventions used restriction, environmental restructuring, modeling and coercion.

The most frequently identified mechanisms of action, by which interventions aimed to change behavior, were ‘psychological capability’ (‘knowledge’, ‘skills’ and ‘behavioral regulation’); ‘reflective motivation’ (‘beliefs about consequences’ and ‘intention’) and ‘physical opportunity’ (‘environmental context and resources, e.g., sharing leaflets with patients); the latter particularly so for community pharmacy interventions. These findings suggest that intervention designers believe that increasing knowledge and motivation among all target groups is key to decreasing inappropriate antibiotic consumption.

Psychological capability was targeted in all groups, delivered largely through ‘instruction on how to perform the behavior. Reflective motivation and psychological capability were targeted in patients and prescribers using the BCT ‘information about health consequences’. Interventions targeting patients and prescribers through ‘beliefs about consequences’ (e.g., NHS England’s Patient Safety Alert [47], Public Health England’s Keep Antibiotics Working campaign [48]) could have drawn on a much wider range of techniques. For example, BCT ‘information about emotional consequences’ (e.g., the anxiety of becoming ill knowing antibiotics would not be effective), or BCT ‘anticipated regret’ (e.g., the degree of regret prescribers may feel in the future if they do not modify their antibiotic prescribing), or by employing incentives or rewards. Provider and commissioner interventions focused more on ‘Feedback on behavior’, for example interventions giving feedback on prescribing and local antimicrobial resistance rates (e.g., NHS England Quality Premium: 2016/17 Guidance for CCGs [49], Public Health England Fingertips platform [50] and PrescQIPP Antimicrobial Stewardship [51]) with the aim of facilitating a change in behavioral regulation (the mental skills to carry out the mental tasks required to perform the behavior).

Physical opportunity was another commonly targeted domain in interventions aimed at all target groups. The most frequently used technique was ‘adding objects to the environment’ (i.e., provision of information leaflets). Many other techniques appropriate for targeting this domain were not used, for example, avoiding/reducing exposure to cues for the behavior which could, e.g., involve diverting patients with self-limiting infections to the pharmacy via appointment booking lines as suggested in Pinder et al. [19].

With the exception of community pharmacy, Interventions were identified across all settings, which targeted social opportunity through social comparison with peers (e.g., UK Chief Medical Officer letter to high prescribers of antibiotics [52]) and practical social support (e.g., Self Care Forum: Self Care Week [53]).

We identified few (8/39) interventions which aimed to change behavior by targeting ‘automatic motivation’, such as routines and habits (e.g., encouraging routine feedback where antibiotics are not prescribed according to guidelines), emotional drives (e.g., addressing concerns about the negative consequences of not prescribing antibiotics), and reinforcement (e.g., incentivizing participation in AMS training), which can be powerful influences on behavior. Designing new interventions and refining existing interventions may merit considering targeting these influences.

## 6. Limitations

There are three key limitations to this study. First, the interventions included in this study are implemented at a national level. The devolution of responsibility to local health and public health teams means the many interventions which were designed and implemented locally are not included. Secondly, we did not establish the extent to which included interventions were effective in changing behavior nor the extent to which they are implemented. Thirdly, whilst we obtained most of the materials and documents related to each included intervention, we were unable to access all materials.

## 7. Future Research Directions

The mechanisms of action of interventions were determined by applying available resources which link COM-B and TDF to intervention content (intervention types and BCTs). This process established how interventions are likely to have an effect on behavior. Future intervention design and refinement would be supported by establishing the barriers and facilitators to the behaviors identified in this study and then comparing them with intervention content to determine the extent to which intervention content appropriately targets identified barriers and facilitators. Establishing the extent to which interventions are effective will support interpretation of these findings. Establishing which behaviors are key in influencing AMR will support the prioritization of intervention design and refinement.

Intervention design and refinement would be aided with the provision of accessible guidance on processes such as that described above to support intervention designers with a range of expertise in behavior change.

## 8. Conclusions

Although changing behaviors alone cannot fully halt the AMR problem, patient and physician behaviors in primary care offer a unique opportunity for AMR interventions. Targeting behaviors amenable to change using effective, evidence-based BCTs, may offer new cost-effective methods to reduce antibiotic prescribing and thus halt the rise in the number of patients suffering from infections due to AMR. This study highlights the need to review existing interventions to ensure they are optimized to influence AMR-related behaviors. Any gaps identified in current provision should be considered for future intervention design and refinement, ensuring these are aligned to work within the NHS’s changing provision of primary care.

## 9. Declarations

### 9.1. Ethics Approval and Consent to Participate

Not applicable.

### 9.2. Consent for Publication

Not applicable.

### 9.3. Availability of Data and Material

The datasets used and/or analyzed during the current study are available from the corresponding author on reasonable request.

## Figures and Tables

**Figure 1 antibiotics-09-00512-f001:**
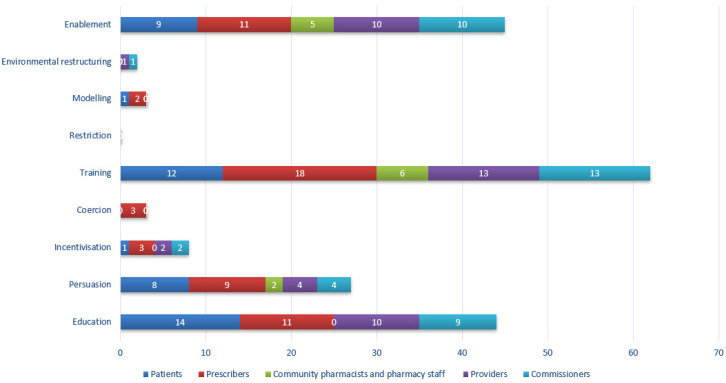
Frequency of identification of intervention types by target group. Total count will exceed the maximum number of interventions (*n* = 39) as many interventions were aimed at more than one target group.

**Figure 2 antibiotics-09-00512-f002:**
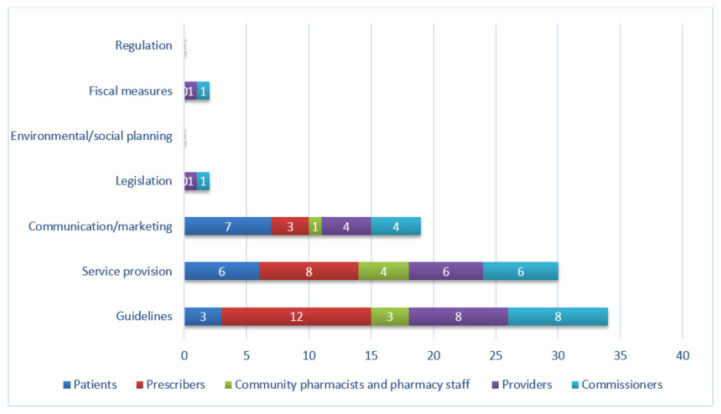
Frequency of identification of policy options, by target group. Total count will exceed the maximum number of interventions (*n* = 39) as many interventions were aimed at more than one target group.

**Figure 3 antibiotics-09-00512-f003:**
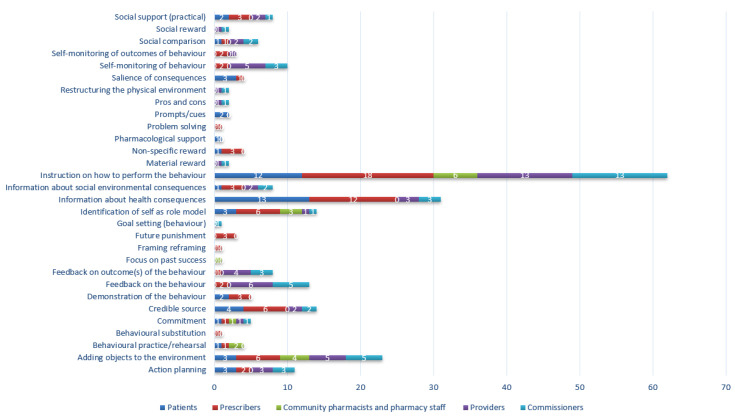
Frequency of identification of BCTs, by target group.

**Figure 4 antibiotics-09-00512-f004:**
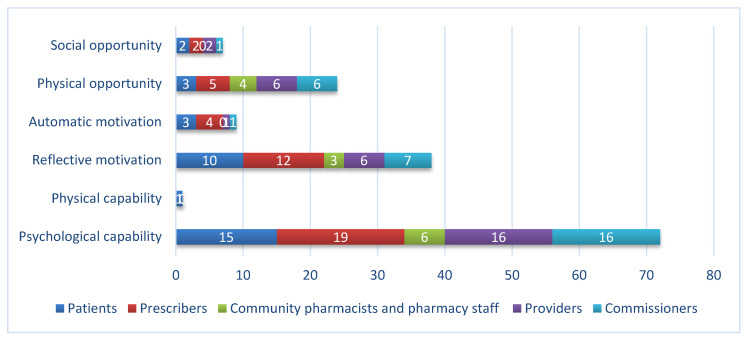
Influences (Capability, Opportunity, Motivation–Behavior (COM-B)) on behavior targeted in interventions, by target group.

**Figure 5 antibiotics-09-00512-f005:**
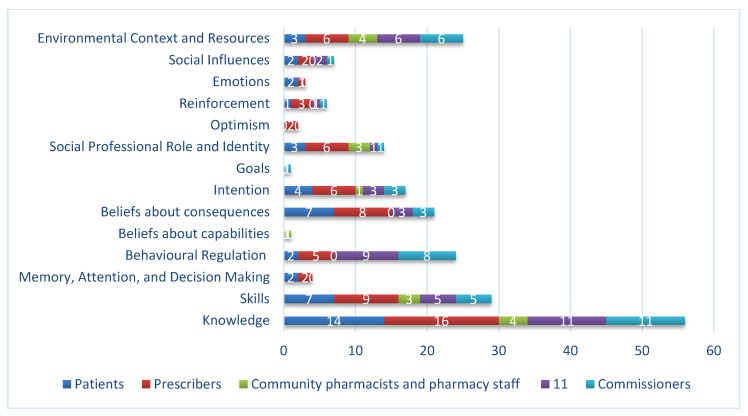
Influences (Theoretical Domains Framework (TDF)) on behavior targeted in interventions, by target group.

**Table 1 antibiotics-09-00512-t001:** Behaviors related to antibiotic prescribing for respiratory tract infections (RTI) in primary care.

Behavior (Number of Behaviors)	Number of Interventions Targeting the Behavior
Patients/Public (and or carers) (*n* = 5)	
Self-care and/or obtain pharmacy advice for signs and symptoms of self-limiting respiratory tract infections prior to, or instead of, a primary care consultation.	13
Do not request antibiotics at primary care consultations for symptoms of self-limiting RTIs.	12
Use back-up prescriptions as directed by a suitably qualified healthcare professional (HCP).	3
Take antibiotics as prescribed (do not save for later use or share with others) by a suitably qualified HCP.	4
Return unwanted antibiotics to the pharmacy.	3
**Primary care prescribers (including non-medical prescribers, such as nurses and pharmacists) *n* = 11**	
Follow/adhere to local antibiotic formulary—general behaviors.	13
Prescribe an antibiotic only when there is likely to be clear clinical benefit, (using fever PAIN or CENTOR for sore throat). OR Do not issue an immediate prescription for an antimicrobial to a patient who is likely to have a self-limiting condition.	19
Give alternative, non-antibiotic self-care advice, where appropriate.	12
Use/share written self-care resources/leaflets when issuing self-care advice for symptoms of self-limiting RTIs.	9
Provide safety netting advice whether or not the patient has been prescribed antibiotics (e.g., what to do if condition gets worse or side effects of medication).	7
When an antibiotic is indicated prescribe the narrowest spectrum antibiotic possible, for the right duration, at the right dose.	9
Provide ‘delayed/back-up’ antibiotic strategy where appropriate.	14
Explain the prescribing decision to the patient, including where appropriate, the benefits and harms of antibiotics.	10
Document, in patients records, clinical diagnosis (including symptoms) if prescribing an immediate or back up antimicrobial and/or giving self-care advice.	6
Undertake POCT in patients 18–65 years old presenting with acute cough/bronchitis in whom antibiotics are being considered.	5
Limit prescribing over the telephone to exceptional cases for self-limiting RTIs.	3
**Community pharmacists and pharmacy staff** **(*n* = 5)**	
Provide self-care advice for patients with symptoms of self-limiting RTIs, instead of, following or prior to referral to a primary care clinician, giving safety netting advice where appropriate.	7
Use/share written resources with the public when providing self-care advice for self-limiting RTIs.	5
When giving an antibiotic prescription for a self-limiting RTI, inform the patients of the dose and duration or to take their antibiotics exactly as prescribed.	3
Check that antibiotic prescriptions comply with local guidance and query with the prescriber for those that do not.	5
Accept and dispose appropriately of returned antibiotics.	2
**Providers and commissioners (*n* = 11)**	
Provide education and training in prudent antimicrobial use/AMR (using the antimicrobial resistance and stewardship competencies as a framework).	6
Commission, develop, or implement interventions (e.g., guidance, services, programs, or campaigns) to support AMS/tackle AMR.	11
Commission, develop, or implement interventions (e.g., guidance, services, programs, or campaigns) to support self-care.	6
Monitor antibiotic prescribing in relation to local and national resistance patterns or targets.	12
Promote current national guidelines, or promote/develop local guidelines on antimicrobial prescribing among all prescribers, providing updates if the guidelines change.	6
Provide regular feedback on antimicrobial prescribing and resistance indicators at prescriber, team, and organization level benchmarked against local or national antimicrobial prescribing/resistance rates.	5
Provide feedback to prescribers on patient safety incidents related to antimicrobial use, including hospital admissions for potentially avoidable life-threatening infections, infections with Clostridium difficile or adverse drug reactions, such as anaphylaxis.	1
Providers have a formulary in place for antibiotic prescribing. *	5
Commissioners seek evidence/providers make evidence available for adherence to local or national guidance for antibiotic prescribing. **	5
Commissioners ensure information and resources are available for healthcare professionals to use during consultations with people seeking advice about managing self-limiting RTIs. **	2
Reduce antibiotic prescribing/antimicrobial resistance—general behaviors.	7

* Providers only.** Commissioners only.

**Table 2 antibiotics-09-00512-t002:** Intervention and target group.

Intervention	Target Group
Public Health England Antibiotic Guardian [37]	Patients
	Prescribers
	Community pharmacists and pharmacy staff
	Providers
	Commissioners
TARGET Antibiotics Toolkit (Treat Antibiotics Responsibly, Guidance, Education, Tools) [38]	Patients
	Prescribers
	Community pharmacists and pharmacy staff
	Providers
	Commissioners
British Society for Antimicrobial Chemotherapy: Antibiotic Action [39]	Patients
	Prescribers
	Community pharmacists and pharmacy staff
	Providers
	Commissioners
Treat Yourself Better [40]	Patients
UK Department of Health and Public Health England Antimicrobial Prescribing and Stewardship Competencies [41]	Prescribers
	Providers
	Commissioners
theLearningpharmacy.com [42]	Community pharmacists and pharmacy staff
Royal College of Nursing (RCN) and Infection Prevention Society (IPS) Infection Prevention and Control Commissioning Toolkit [43]	Providers
Stemming the Tide of Antibiotic Resistance (STAR) e-learning [44]	Prescribers
Managing Acute Respiratory Tract Infections (MARTI) e-learning [45]	Prescribers
Public Health England ‘Beat the Bugs’ course [46]	Patients
NHS England Patient Safety Alert—addressing antimicrobial resistance through implementation of an antimicrobial stewardship program [47]	Providers
	Commissioners
Public Health England Keep Antibiotics Working campaign [48]	Patients
	Providers
	Commissioners
NHS England Quality Premium: 2016/17 Guidance for CCGs [49]	Providers
	Commissioners
Public Health England Fingertips platform [50]	Providers
	Commissioners
PrescQIPP Antimicrobial Stewardship [51]	Prescribers
	Providers
	Commissioners
UK Chief Medical Officer letter to high prescribers of antibiotics [52]	Prescribers
Self Care Forum: Self Care Week [53]	Patients
	Providers
	Commissioners
The Health and Social Care Act (HSCA) 2008. Code of Practice on the prevention and control of infections and related guidance [54]	Providers
	Commissioners
NICE Antimicrobial stewardship: systems and processes for effective antimicrobial medicine use [NG15] [17]	Prescribers
	Community pharmacists and pharmacy staff
	Providers
	Commissioners
NICE Infection Prevention and Control [QS61] [27]	Patients
	Prescribers
	Providers
	Commissioners
UK Five Year Antimicrobial Resistance Strategy 2013 to 2018 [15]	Prescribers
	Providers
	Commissioners
NHS website advice on common cold [55]	Patients
NICE Antimicrobial stewardship: changing risk-related behaviors in the general population [NG63] [23]	Prescribers
	Community pharmacists and pharmacy staff
	Providers
	Commissioners
Center for Pharmacy Postgraduate Education distance course: Antibacterial resistance—a global threat to public health: the role of the pharmacy team [56]	Prescribers
	Community pharmacists and pharmacy staff
UK Clinical Pharmacy Association / Royal Pharmaceutical Society—professional practice curriculum [57]	Prescribers
FeverPAIN [58]	Prescribers
Public health England Managing Common Infections Guidance [28]	Prescribers
NICE Respiratory tract infections (self-limiting): prescribing antibiotics [CG69] [59]	Patients
	Prescribers
NICE Antimicrobial Stewardship [QS121] [29]	Prescribers
	Providers
	Commissioners
Self Care Forum: Factsheet 7 (Cough in Adults); Factsheet 12 (Common Cold) [60]	Patients
OpenPrescribing.net [61]	Providers
	Commissioners
CENTOR [62]	Prescribers
Health Education England ‘Antimicrobial Resistance: A Guide for GPs’ [63]	Prescribers
NICE Sinusitis (acute): antimicrobial prescribing [NG79] [24]	Patients
	Prescribers
NICE Sore throat (acute): antimicrobial prescribing [NG84] [26]	Prescribers
Department of Health & Social Care ‘Take Care not Antibiotics’ videos [64]	Patients
Patient.info webpages on colds, sore throats, antibiotics, bronchitis and sinusitis [65]	Patients
Health Education England ‘Awareness of Antimicrobial Resistance (AMR) Animation’ [66]	Patients
Royal Pharmaceutical Society: Antimicrobial Stewardship Quick Reference Guide [31]	Community pharmacists and pharmacy staff
	Commissioners

**Table 3 antibiotics-09-00512-t003:** Interrater reliability.

	Kappa	PABAK
**Behaviors**	0.89	0.90
**COM-B**	0.68	0.75
**TDF**	0.60	0.77
**Intervention types**	0.67	0.76
**Policy options**	0.77	0.88
**BCTs**	0.69	0.95

**Table 4 antibiotics-09-00512-t004:** Frequency of types in interventions.

Intervention Type	Number of Interventions (Max 39)	Example of Coded Intervention
Training	32	“Ensure resources and advice are also available for people who are prescribed or supplied with antimicrobials, to ensure they take them as instructed by their healthcare professional. This should include taking the correct dose for the time specified and via the correct route.”
Education	29	“Taking antibiotics encourages harmful bacteria that live inside you to become resistant.”
Enablement	25	Resources listed to become an Antibiotic Guardian.
Persuasion	18	“Once the bugs are resistant, the antibiotics don’t work and we’re back in the ‘the Stone Age.’”
Incentivization	6	CPD points for completing a course.
Modeling	3	Film of a consultation where a physician manages a patient’s expectations.
Coercion	3	“If we don’t act now, any one of us could go into hospital in 20 years for minor surgery and die because of an ordinary infection that can’t be treated by antibiotics.”
Environmental restructuring	1	“Service providers (such as hospitals and dental practices) ensure that prescribers of antimicrobials have access to electronic prescribing systems that link indication with the antimicrobial prescription.”
Restriction	0	-

**Table 5 antibiotics-09-00512-t005:** Frequency of Behavior Change Techniques (BCTs) in interventions.

BCT	Number of Interventions Identified in (Max 39)	Example of BCT Identified in an Intervention
Instruction on how to perform the behavior	33	“Rest, drink plenty of fluids, take pain relievers, such as paracetamol or ibuprofen, and talk to your pharmacist for advice on getting the relief you need.”
Information about health consequences	25	“Most common winter ailments, such as colds, sore throat, cough, sinusitis, or painful middle ear infection (earache), can’t be treated with antibiotics.”
Adding objects to the environment	11	Providing decision aids for antibiotic prescription.
Credible source	11	Letter from the Chief Medical Officer.
Action planning	8	Provision of an implementation spreadsheet in Antibiotic stewardship Quality Standard QS121.
Feedback on the behavior	8	Comparative data on national and local prescribing.
Identification of self as role model	8	“All pharmacists, regardless of setting, have AMS obligations and with over 1.6 million visits each day, community pharmacy teams have a key role.”
Information about social environmental consequences	6	“Antibiotic prescribing is a huge cost for the NHS. For instance annual prescribing for acute cough alone exceeds £15 million (NICE, 2008).”
Feedback on outcome(s) of the behavior	6	National and local data on prescribing outcomes.
Social support (practical)	5	“If you are not sure, ask your doctor, nurse practitioner or pharmacist for help and advice.”
Social comparison	4	“Ranking - Looks at how you compare, and rank, against the 211 CCGs nationally, your 10 ‘cluster’ CCGs (most like you), and within the PrescQIPP Community. We also provide the Range within these groups”
Demonstration of the behavior	4	Video clip of a consultation showing GPs explaining why antibiotics are not useful to treat a virus.
Salience of consequences	4	“If no antibiotics work, it will be like going back to the 1930s.”
Self-monitoring of behavior	4	“Review and monitor how well the guideline is being implemented through the project group.”
Future punishment	3	“The rapid spread of multi-drug resistant (MDR) bacteria means that we could be close to reaching a point where we may not be able to prevent or treat everyday infections or diseases.”
Self-monitoring of outcomes of behavior	3	Testing GPs knowledge on completion of an RTI self-management training module.
Non-specific reward	2	“Certificate for course completion.”
Prompts/cues	2	‘Choose self-care for life’ posters/web buttons/TV screens.
Material reward	1	“Reward for improvements in service quality.”
Behavioral practice/rehearsal	1	“Rehearsal and discussion of what to do when have cough or cold”
Behavioral substitution	1	Recommending providing patients with a leaflet on UTI management rather than prescribing antibiotics.
Commitment	1	Making a pledge, e.g., “I will ensure all prescribers in my practice including locums have easy access to the local antibiotic guidance.”
Focus on past success	1	Reminding pharmacists about their experience in providing self-care advice.
Framing reframing	1	Suggesting that leaflets are viewed as a tool for self-management rather than a parting gift.
Goal setting (behavior)	1	“Quality standards are intended to drive up the quality of care, and so achievement levels of 100% should be aspired to.”
Pharmacological support	1	“Talk to your local pharmacist about other ways to help with symptoms, such as taking painkillers.”
Problem solving	1	“Identifying barriers to change in prescribers own practice.”
Pros and cons	1	Conducting a SWOT analysis (Strengths, Weaknesses, Opportunities, Threats) in planning implementing a patients self-care intervention.
Restructuring the physical environment	1	“Electronic prescribing systems that link indication with the antimicrobial prescription.”
Social reward	1	“Thank you for your ongoing commitment to reduce antimicrobial resistance (AMR) and drug resistance infections.”

**Table 6 antibiotics-09-00512-t006:** Frequency of interventions targeting influences on behavior.

	Number of Interventions (Max 39)
**COM-B**	
Psychological capability	38
Reflective motivation	22
Physical opportunity	12
Automatic motivation	8
Social opportunity	6
Physical capability	1
**TDF**	
Knowledge	33
Skills	16
Beliefs about consequences	15
Environmental Context and Resources	13
Behavioral Regulation	12
Intention	11
Social Professional Role and Identity	7
Social Influences	6
Reinforcement	5
Memory, Attention, and Decision Making	4
Emotion	3
Optimism	2
Beliefs about capabilities	1
Goals	1

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
