# Peer review of "Content and Mechanism of Action of National Antimicrobial Stewardship Interventions on Management of Respiratory Tract Infections in Primary and Community Care"

_antibiotics, 2020, doi:10.3390/antibiotics9080512_

Round 1
Reviewer 1 Report
Increasing antibiotic resistance among pathogenic bacteria is one of the most serious public health threats. Since antibiotic overuse is the main factor promoting the emergence of antimicrobial resistance, there is a need to develop efficient Antibiotic stewardship guidelines and policies. In the presented manuscript Atkins and colleagues used the Behavioral Change Wheel, a method for designing behavior change interventions, to analyze antibiotic-stewardship programs in England. The Authors concluded that most interventions focused on education and training through the provision of instructions and information about health consequences. The Authors also indicated relevant Behavioral change techniques that may improve the interventions. This study is comprehensive and well-documented. I have only minor comments:
Lines 137-138 - the sentence should be corrected,
Page 21, Conclusions, the last long sentence seems to be unclear.
Page 26 – Please, add “Additional file 4” to the list.
Line 171 –Please explain the abbreviation “BCTTv1”
Author Response
Please find below our responses to the Reviewer's helpful comments.
Lines 137-138 - the sentence should be corrected
We have amended this in line with this and Reviewer 2’s comments ““We drew on the following sources to identify behaviours relevant to patient self-care and/or appropriate antibiotic advice for the management of signs and symptoms of self-limiting respiratory tract infections”
Page 21, Conclusions, the last long sentence seems to be unclear.
We have amended this to “This study highlights the need to review existing interventions to ensure they are optimised to influence AMR-related behaviours. Any gaps identified in current provision should be considered for future intervention design and refinement , ensuring these are aligned to work within the NHS’s changing provision of primary care.”
Page 26 – Please, add “Additional file 4” to the list.
We have added Additional file 4 to the list of additional files.
Line 171 –Please explain the abbreviation “BCTTv1”
We have added text to explain this abbreviation where it is first mentioned in line 118 “Behaviour Change Technique (BCT) Taxonomy version 1 (BCTTv1)”
Reviewer 2 Report
Thank you very much for inviting me to review the manuscript. The manuscript entitled “Content and mechanism of action of national antimicrobial stewardship interventions on management of respiratory tract infections in primary and community care” highlights the misuse of antibiotics and describes the action of AMS interventions on the management of respiratory tract infections in primary and community care.
The manuscript is well written; however, there are many long, incomplete, and confusing sentences. The authors must revise the following sentences, not limited to…
Line 24-27
Line 29-33: Repetitious of words like “and”
Line 35: Avoid starting a paragraph/sentence with a number. This is not wrong; however, it is untidy.
Line 39-46. Revise these sentences.
Line 52 to 53: Revise
Line 61-63: Revise
Line 71-73: Revise
Line 81-82: Revise
Line 145-148: Revise
Line 155-166; Revise
-There are many other long and poorly constructed sentences that need to be corrected. Please thoroughly check the entire manuscript to avoid repetition and redundancy.
-Page 14: BCTs identified in interventions
-The first two sentences could be merged as “ A total of 30 BCTs were identified across all interventions with the mean number of 4 per intervention (range 1-14) as shown in table 5”.
-The third sentence: Patient …..often: must be revised.
-Remove the period from the title
-Besides, the authors should explain in detail the “mechanism of action” and discuss it concerning current literature.
-Overall, it is an interesting manuscript. However, language needs more polishing to make it easy for readers to understand. The majority of the sentences are confusing, and some are incomplete.
Author Response
We thank the reviewer for these helpful comments to simplify the manuscript text and provide the following responses.
Line 24-27
Amended to “This study aimed to describe the content and mechanism of action of antimicrobial stewardship (AMS) interventions to improve appropriate antibiotic use for respiratory tract infections (RTI) in primary and community care. This study also aimed to describe who these interventions were aimed at and the specific behaviours targeted for change”
Line 29-33: Repetitious of words like “and”
Amended to “Evidence-based guidelines, peer-review publications and infection experts were consulted to identify behaviours relevant to AMS for RTI in primary care and interventions to target these behaviours. Behaviour change tools were used to describe the content of interventions. Theoretical frameworks were used to describe mechanisms of action”
Line 35: Avoid starting a paragraph/sentence with a number. This is not wrong; however, it is untidy.
Amended to “A total of 32 behaviours targeting six different groups were identified”
Line 39-46. Revise these sentences.
Amended to “Influences on behaviour most frequently targeted by interventions were psychological capability (knowledge and skills); reflective motivation (beliefs about consequences, intentions, social/professional role and identity) and physical opportunity (environmental context and resources). Interventions were most commonly characterized as achieving change by training, enabling or educating and were delivered mainly through guidelines, service provision and communication & marketing. Interventions included a mean of four BCTs (range 1-14). We identified little intervention content targeting automatic motivation and social opportunity influences on behaviour”
Line 52 to 53: Revise
Amended to “This study provides a platform to refine content and plan evaluation of antimicrobial stewardship interventions”
Line 61-63: Revise
Amended to “Appropriate antibiotic use is key to addressing antimicrobial resistance. It is currently unclear which behaviours are targeted by antimicrobial stewardship interventions. The mechanisms of action of these interventions also require greater articulation”
Line 71-73: Revise
Amended to “It is estimated that a continued rise in resistance would cost the world 100 trillion USD by 2050 if AMR is not addressed effectively”
Line 81-82: Revise
Amended to “The first step in intervention design is to specify the behaviour(s) the intervention is aimed at changing”
Line 145-148: Revise
We have merged this with the first line of the methods to be more concise “We drew on the following sources to identify behaviours relevant to patient self-care and/or appropriate antibiotic advice for the management of signs and symptoms of self-limiting respiratory tract infections”
Line 155-166; Revise
Amended to “We included interventions implemented nationally in England between January 2014 and February 2018, where the primary objective was antimicrobial stewardship activities related to patient self-care and/or appropriate antibiotic advice for the management of respiratory tract infections. Interventions were excluded if local implementation only had occurred. Two authors (AS & VL) sought descriptions of the interventions from publicly available material and in some instances contacted the programme leads for elaboration and to ensure accurate descriptions”
-There are many other long and poorly constructed sentences that need to be corrected. Please thoroughly check the entire manuscript to avoid repetition and redundancy.
-Page 14: BCTs identified in interventions
-The first two sentences could be merged as “ A total of 30 BCTs were identified across all interventions with the mean number of 4 per intervention (range 1-14) as shown in table 5”.
Amended as suggested by Reviewer 2.
-The third sentence: Patient …..often: must be revised.
Amended to “Patient and prescriber interventions most commonly used the BCTs ‘information about health consequences’ and ‘instruction on how to perform the behaviour’. The most frequently used BCT in pharmacy, provider and commissioner interventions was ‘instruction on how to perform the behaviour’”
-Remove the period from the title
We have removed this.
-Besides, the authors should explain in detail the “mechanism of action” and discuss it concerning current literature.
Paragraphs 4-8 of the discussion describe which mechanisms of action were targeted in interventions and the BCTs used to target them.
-Overall, it is an interesting manuscript. However, language needs more polishing to make it easy for readers to understand. The majority of the sentences are confusing, and some are incomplete.
We have made the following amendments to reduce redundancy of language:
- Line 129 “Multifaceted AMS programmes aimed at the public as well as frontline healthcare professionals are needed to tackle AMR [16, 17]”
- Line 140 “To optimise the potential of AMS programmes, the behaviours and populations targeted by these interventions as well as their content and mechanisms of action of these needs to be articulated”
- Line 170 “The overarching aim of this study was to build on the work of Pinder et al. [19] by characterising nationally implemented AMS primary care programmes to identify any gaps in coverage and opportunities for refinement”
- Discussion, paragraph 2 – the second sentence which was four lines long has been divided into two sentences.
- Discussion, paragraph 5 – the third sentence has been divided into two sentences.
Round 2
Reviewer 2 Report
The revised manuscript could be accepted for its possible publication in "Antibiotics"